# Developing Soybean Protein Gel-Based Foods from Okara Using the Wet-Type Grinder Method

**DOI:** 10.3390/foods10020348

**Published:** 2021-02-06

**Authors:** Yuya Arai, Katsuyoshi Nishinari, Takao Nagano

**Affiliations:** 1Department of Food Science, Faculty of Bioresources and Environmental Sciences, Ishikawa Prefectural University, 1-308, Suematsu, Nonoich 921-8836, Japan; evian0429akyg@gmail.com; 2Glyn O. Phillips Hydrocolloids Research Centre, School of Food and Biological Engineering, Hubei University of Technology, Wuhan 430068, China; katsuyoshi.nishinari@gmail.com

**Keywords:** okara, wet-type grinder, nanocelluloses, soy protein isolate, heat-induced gel

## Abstract

Okara, a by-product of tofu or soymilk, is rich in dietary fibers (DFs) that are mostly insoluble. A wet-type grinder (WG) system was used to produce nanocellulose (NC). We hypothesized that the WG system would increase the dispersion performance and viscosity of okara. These properties of WG-treated okara improve the gel-forming ability of soybean proteins. Here, the suspensions of 2 wt% okara were treated with WG for different passages (1, 3, and 5 times). The particle size distribution (PSD) and viscosity of WG-treated okara decreased and increased, respectively, with different passages. The five-time WG-treated okara homogeneously dispersed in water after 24 h, whereas untreated okara did not. The breaking stress, strain, and water holding capacity of soybean protein isolate (SPI) gels increased upon the addition of WG-treated okara. This effect increased as the number of WG treatments increased. The breaking stress and strain of SPI gels to which different concentrations of the five-time WG-treated okara were added also increased with increasing concentrations of WG-treated okara. These results suggest that NC technology can improve the physicochemical properties of okara and are useful in the development of protein gel-based foods.

## 1. Introduction

Okara is a by-product of tofu or soymilk and rich in insoluble dietary fibers (DFs), which mostly consist of cellulose and hemicellulose [1,2,3]. Thus, improving the physicochemical properties of okara for various food applications is of great importance [4,5,6,7,8]. In previous studies, Huang et al. [4] modified okara using enzyme, chemical, and homogenization methods. All treatments improved the swelling capacity of okara, and the homogenization and enzyme-treatment were the most appropriate methods to improve the solubility of okara [4]. Dai et al. [9] also investigated the effects of modified okara consumption on body weight, serum metabolites, and fatty acid compositions in high fat-fed mice. Intake of the modified okara suppressed body weight gain, changes in metabolites, and fatty acid compositions in mouse sera compared to untreated mice [9]. Pérez-López et al. [5] studied the treatment with high hydrostatic pressure (HHP) and food-grade enzymes to increase the amount of soluble DF in okara. HHP and enzyme treatment increased the amount of soluble DF up to 13.53 ± 0.3% [5]. Additionally, the effects of dietary HHP and enzyme-treated okara were examined in high fat-fed mice. HHP and enzyme-treated okara intake inhibited the increase in body and liver weights and controlled triglycerides and urea in plasma [10]. Ullah et al. [6] examined okara by high-energy wet milling and reduced the particle size of okara from 66.7 μm to 544.3 nm. This research group also studied the effect of nanosized okara on the gel properties of tofu [11] and silver carp surimi [12]. These studies demonstrated that pulverization can improve the physicochemical and functional properties of okara. However, the usefulness of pulverized okara in food applications is unclear.

Water jet (WJ) and wet-type grinder (WG) systems are commonly used to produce nanocelluloses (NCs) [13,14,15]. The WJ system involves a high-pressure water jet technology that can pulverize sample suspensions in water up to 245 MPa. With the WJ-treatment at 245 MPa in ten cycles, the crystalline cellulose (CC) suspended in water (1% *w/v*) was delaminated, which indicates a highly viscous and a gel-like form. Scanning transmission electron microscopy images showed that width of the WJ-treated CC fiber was reduced to 25 nm, indicating the formation of NC. This system also decreased the crystal structure of cellulose and tripled the hydrolysis efficiency of cellulase [16]. In our previous study, we used a WJ system, pulverized okara, and microcrystalline cellulose (MCC). The WJ-treated okara and MCC had high dispersion performance and viscosity. The WJ-treated okara also showed greater inhibition of α-amylase compared to that presented by MCC and cellulose. Furthermore, WJ-treated okara increased butyrate production by *Roseburia intestinalis*, which is one of the dominant species of human gut microbiota [7]. In our other studies, we investigated the effects of dietary NC produced by the WJ system on obesity and the gut microbiota of mice fed a high-fat diet [17,18]. NC consumption suppressed weight gain and fat accumulation [17]. The consumption of NC increased voluntary wheel-running activity, inhibited the increase in body weight, and controlled the balance of gut microbiota in mice [18]. Ifuku et al. reported methods for the preparation of chitin and chitosan nanofibers (CNFs) using the WJ system [19,20]. The authors also demonstrated the beneficial effects of CNFs as functional foods in animal models [21]. These results show that the WJ system is a useful method to not only produce NC, but also to develop ingredients for functional foods. However, to process samples with the WJ system, they should have a particle size of less than 0.1 mm. In contrast, the WG system passes the cellulose dispersion in water between two grinding stone disks, where the distance between the two disks can be adjusted. The zero position between the two disks is the contact position, and a negative gap between the two disks can be set under wet conditions to increase the fibrillation efficiency. An advantage of this system is its ability to avoid clogging that often occurs in a high-pressure homogenizer and the WJ system [22,23,24]. These reports indicate that the WG system is more applicable than the WJ system in the food industry.

Processed soybean foods, such as tofu and soymilk, are often present in human diets. However, these products contain low DFs, although soybeans themselves contain a high percentage of DFs (15.6–17.9%). For this reason, there are some reports where tofu has been prepared from whole soybeans, but not soymilk [25]. In addition, the replacement of animal proteins with plant proteins has been driven by the concept of sustainability assurance in food production. Soybean-based ingredients are used in many meat analog products currently in the market [26,27,28]. Therefore, the gel-forming ability of soybean proteins is the most important property for the development of processed soybean foods [29,30].

We hypothesized that the WG system would increase the dispersion performance and viscosity of okara. These properties of WG-treated okara improve the gel-forming ability of soybean proteins. To test this hypothesis, the aim of this study was to explore the effect of WG-treated okara on the gel-forming ability of soybean proteins. We examined (1) okara pulverization using the WG system and (2) the effects of WG-treated okara on soybean protein isolate (SPI) gel properties.

## 2. Materials and Methods

### 2.1. Materials

Defatted okara (Newproplus 1000) and SPI (Fujipro F) were obtained from Fuji Oil Co., Ltd. (Izumisano, Japan). According to the manufacturer, Newproplus 1000 contains 6.0% water, 20.7% protein, 0.2% fat, 69.1% carbohydrate (63.7% dietary fiber), and 4.0% ash. Fujipro F contains 5.0% water, 86.3% protein, 0.2% fat, 4.0% carbohydrate, and 4.5% ash.

### 2.2. Preparation of WG-Treated Okara

Dispersion of 2 wt% okara in distilled water was pulverized using a Supermasscolloider (MKCA6-2; Masuko Sangyo Co., Ltd., Kawaguchi, Japan) with a −0.15 mm gap at 1540 rpm of the stone disk.

### 2.3. Viscosity of WG-Treated Okara

The viscosity was measured using a B-type viscometer (Toyo Keiki Inc., Tokyo, Japan) with a No. 1 rotor at a shear rate of 0.5 s^−1^ at 25 °C [1]. The data represent the average of three measurements for each sample.

### 2.4. Particle Size Distribution (PSD) of WG-Treated Okara

The particle size distribution (PSD) was measured using a laser scattering PSD analyzer (LA-960, Horiba, Kyoto, Japan) with a relative refractive index of 1.60 [7]. The data represent the average of two measurements for each sample.

### 2.5. Preparation of SPI Gel

SPI gels were prepared as described in our previous study [31]. SPI powder (6 wt%) was hydrated by mixing with a WG-treated okara slurry and dispersed using a homogenizer (IKA Ultra-Turrax T8 Disperser, IKA Works, Inc., Staufen im Breisgau, Germany). After adding 0.25% magnesium chloride (MgCl_2_) and sodium chloride (NaCl), the slurry was poured into a stainless mold (inner diameter 20 mm and height 20 mm), whose inside surface was covered with silicon grease. A silicon rubber sheet and a polycarbonate plate (25 × 25 mm) were placed on both the top and bottom of the mold and then were tightly sealed using two rubber bands. The molds were heated at 80 °C for 30 min in a water bath (FSGPD05, Fisher Scientific International Inc., Hampton, NH, USA).

### 2.6. Compression Measurements of SPI Gels Containing WG-Treated Okara

Compression measurements were performed using a texture analyzer (TA-XT2iHR, Stable Micro Systems) attached to a 5 kg load cell at 25 °C. A cylindrical plunger with a diameter of 50 mm was used and the compression speed was 1 mm/s. In each experiment, at least six gels were examined for each point.

### 2.7. Water Holding Capacity (WHC) of SPI Gels Containing WG-Treated Okara

The sample mixtures were prepared as described in Section 2.5. Briefly, SPI was mixed with a WG-treated okara slurry and 0.25% MgCl_2_ and 1% NaCl were added. Water content (W) and water holding capacity (WHC) were measured using a modification of a previously reported method [32]. An aluminum cup was weighed (*m*_0_) and the sample mixture was added and then weighed again (*m*_1_). The cup was heated using a drying oven at 105 °C overnight, cooled to 25 °C in a desiccator, and then weighed (*m*_2_). The W was calculated using Equation (1):(1)$$W\;=\;1\;–\;\frac{m2\;–\;m0}{m1\;–\;m0}$$

A 1.5 mL microtube was weighed (*m_3_*) and sample mixtures were added and then weighed (*m*_4_). The gel was formed by heating at 80 °C for 30 min and the tube was centrifuged at 1000× *g* for 10 min. After centrifugation, the upper phase was weighed (*m*_5_). WHC was calculated using the following Equation (2):(2)$$\mathrm{WHC}\;\left(\%\right)\;=\;100\frac{\left(m4\;–\;m3\right)\;W\;–\;m5}{\left(m4\;–\;m3\right)\;W}$$

The data represent the average of three measurements for each sample.

### 2.8. Statistical Analyses

Data represent mean ± standard deviation (SD). Statistical significance was calculated using one-way analysis of variance (ANOVA), followed by Tukey’s post-hoc test, using the Origin 2020b software (Origin Lab, Northampton, MA, USA). Data were considered statistically significant at *p* < 0.05.

## 3. Results

### 3.1. PSD, Viscosity, and Dispersion Ability of WG-Treated Okara

To investigate the impact of WG system on okara properties, we examined PSD, viscosity, and dispersion ability. The dispersions of 2 wt% okara in water were treated with the WG system after different passages (1, 3, and 5 times). The PSD median sizes of okara decreased with increasing passages. In contrast, the okara viscosity increased with increasing passages (Table 1). The WG-treated okara was dispersed homogeneously in water with increasing passages (1, 3, and 5 times) after 24 h (Figure 1). These results indicate that the viscosity and dispersion ability of WG-treated okara increased with increasing number of passages.

### 3.2. Effects of NaCl Concentrations on SPI Gels

We examined the effect of NaCl concentration on the rheological properties of 6% SPI gels, because NaCl was required for the gel formation. We previously described the pronounced effect of NaCl concentration on SPI gel properties [33]. The breaking stress of the SPI gels at 0.2%, 0.5%, 1.0%, and 1.5% NaCl showed values of 160 ± 20 Pa, 400 ± 40 Pa, 270 ± 20 Pa, and 190 ± 25 Pa, respectively (Figure 2a), while the values for the breaking strain of these SPI gels were 32.9 ± 2.0%, 42.0 ± 1.5%, 37.0 ± 0.7%, and 34.0 ± 2.3%, respectively (Figure 2b). The breaking stress and strain of the SPI gels increased with an increase in the NaCl concentration up to 0.5% and then decreased.

### 3.3. Effects of WG-Treated Okara, with Varying Passages, on SPI Gels

To clarify the effect of WG-treated okara with different passages in the WG system on SPI gels, we studied the gel properties of SPI at 1.0% and 1.5% NaCl (Figure 3). We examined the breaking stress and strain of 6% SPI gels at 1.0% NaCl with 1 wt% okara treated with WG after different passages. The breaking stress of SPI gels to which untreated okara and WG-treated okara after one, three, and five passages were added and showed values of 365 ± 43 Pa, 360 ± 32 Pa, 495 ± 70 Pa, and 529 ± 66 Pa, respectively (Figure 3a), while the breaking strain of these SPI gels was 40.4 ± 2.1%, 40.9 ± 0.7%, 43.6 ± 1.9%, and 44.0 ± 0.6%, respectively (Figure 3b). The breaking stress and strain of SPI gels containing three- and five-time WG-treated okara were significantly higher than those of gels with untreated okara (*p* < 0.01).

We also measured the breaking stress and strain of 6% SPI gels at 1.5% NaCl concentration with 1 wt% okara treated with WG after varying the number of passages. The breaking stress of SPI gels, to which untreated okara and WG-treated okara after one, three, and five passages were added, showed values of 190 ± 8 Pa, 200 ± 10 Pa, 270 ± 9 Pa, and 360 ± 35 Pa, respectively (Figure 3c), while the values for the breaking strain of these SPI gels were 34.2 ± 1.1%, 33.5 ± 0.7%, 37.1 ± 0.6%, and 37.6 ± 2.2%, respectively (Figure 3d). The breaking stress and strain of the SPI gels containing three and five-time WG-treated okara were significantly higher than those of the gels with untreated okara (*p* < 0.01).

We determined the WHC of 6% SPI gels at 1.0% NaCl containing 1 wt% okara treated with WG after different passages. The WHC of SPI gels after the addition of untreated okara and WG-treated okara following one, three, and five passages were 93.30 ± 0.75%, 94.12 ± 1.05%, 95.20 ± 0.97%, and 96.16 ± 0.87%, respectively (Figure 4). The WHC of SPI gels consisting of three- and five-time WG-treated okara were significantly higher than those of SPI gels with untreated okara (*p* < 0.01).

### 3.4. Effects of WG-Treated Okara Concentration on SPI Gels

To clarify the effect of WG-treated okara concentration on SPI gels, we studied the gel properties using compression measurements. We determined the breaking stress and strain of 6% SPI gels to which different concentrations of the five-time WG-treated okara were added at 1.0% NaCl. The breaking stress of SPI gels after addition of 0%, 0.5%, 0.75%, and 1% WG-treated okara were 257 ± 22 Pa, 412 ± 15 Pa, 513 ± 34 Pa, and 531 ± 61 Pa, respectively (Figure 5a), whereas the breaking strain of these SPI gels were 36.0 ± 0.9%, 39.1 ± 0.4%, 40.7 ± 0.8%, and 42.0 ± 2.5%, respectively (Figure 5b). The breaking stress and strain of SPI gels significantly increased with increasing concentration of WG-treated okara (*p* < 0.01).

## 4. Discussion

The results of this study show that WG can be a useful technology for improving the physicochemical properties of okara, such as dispersion ability and viscosity. In addition, the WG-treated okara improved SPI gel properties, such as breaking stress and strain, as well as WHC.

In the present study, the median PSD size decreased with increasing passages up to the third time, and there was little difference in the median sizes of WG-treated okara after three and five passages. The PSD median sizes were reduced to 8.9 μm by WG treatment after five passages. The viscosity and dispersion performance after 24 h of WG treatment of okara increased with an increase in the number of passages. In our previous study, we had used the WJ system to improve the physicochemical properties of okara. The dry grinder (DG) treatment reduced the PSD median sizes of okara from 61.5 μm to 39.7 μm and the WJ system decreased its size to 6.6 μm. We determined the dispersion performance by placing the samples for 24 h. The untreated and DG-treated okara dispersions produced a precipitate after 24 h, whereas the WJ-treated okara did not precipitate [7]. These results indicate that okara can be enhanced by both systems in terms of the viscosity and dispersion ability, although the WJ system atomizes okara more efficiently than the WG system. Additionally, Iwamoto et al. [23] investigated pulp fiber delamination using the WG system. The images of the scanning electron microscope showed many micro-sized fibers by one passage, and the size of the fibers reduced with further treatments. Most fibers became nano-sized after five passages; however, further passages did not change the fiber morphology. Our present results are in line with those obtained in the study by Iwamoto et al. [23].

In this study, MgCl_2_, used in tofu production, was used as a coagulant. The heat-induced SPI gels did not form without the addition of NaCl, and the breaking stress and strain of the SPI gels increased up to a 0.5% NaCl concentration and then decreased. These results show that ionic interactions are important for the formation and texture of SPI gels. This phenomenon could be due to ionic repulsion and attraction, which decrease with an increase in the ionic strength of the system [30]. 

Our study showed that the addition of WG-treated okara increased the breaking stress and strain and WHC of SPI gels. This effect increased as the number of WG treatments increased. The breaking stress and strain of WG-treated okara and SPI gels also increased with an increase in concentrations of five-time WG-treated okara. In a previous study, Liu et al. [25] applied ultra-high-pressure homogenization (UHPH) to soybean flour for tofu making. UHPH reduced the particle size of soybean flour and the hardness of tofu prepared from UHPH-treated soybean flour was similar to that of the control tofu. In other previous studies, Ullah et al. [11] studied the effect of nano-sized (370 nm) and micro-sized (110 μm) okara on the gel properties of tofu. Soymilk was mixed with different volumetric ratios of nano-sized and micro-sized okara suspensions. Nano-sized okara tofu showed more whiteness than micro-sized okara tofu. Nano-sized okara was also distributed well in the gel matrices and provided less gritty mouthfeel than micro-sized okara. This research group also investigated the effects of nano-sized okara on the gel properties of silver carp surimi. The breaking force and penetration distance of the surimi gels increased with increasing concentration of nano-sized okara. Light microscopy images showed that nano-sized okara, but not micro-sized okara, distributed well in surimi gels [12]. These results suggest that dehydration and filler effects are possible reasons for controlling gel properties. The increased WHC of SPI gels could be attributed to the high water absorption ability of WG-treated okara. The higher water absorption ability corresponds to a higher dehydration effect in the gel matrix. The smaller particle size of okara generated by WG treatments could remain in the gel matrix without interrupting it, which corresponds to the filler effects. However, the underlying mechanism for this could not be clarified in this study and thus might warrant further investigation.

In processed surimi and meat products, the NaCl concentration ranges from 1% to 3% [34,35]. In this study, the breaking stress and strain of SPI gels were improved by adding WG-treated okara at NaCl concentrations of 1.0% and 1.5%. This effect increased by increasing the WG-treated okara concentrations in the range 0.5%–1.0%. The WHC was also improved by the addition of WG-treated okara. Our findings suggest that WG-treated okara improves processed food products. Meanwhile, the effects of DFs and resistant starch on protein gels of surimi and meat have been investigated for added health benefits [36,37,38]. Alakhrash et al. [36] studied the effects of oat bran on the physicochemical properties of Alaska pollock surimi gels. Their investigation demonstrated that incorporating oat bran into surimi products did not compromise quality [36]. In another study on surimi, Yang et al. [37] investigated the effects of a highly resistant rice starch (RS) on the properties of surimi gels from grass carp. The optimum RS addition level was proposed to be 4% *w*/*w* [37]. Additionally, in a study on meat products, Zhao et al. [38] studied the effects of regenerated cellulose (RC) fibers on myofibrillar protein gels from lean pork. They showed that the textural properties and WHC of myofibrillar protein gels were enhanced with the RC fibers [38]. These studies could contribute to the development of protein-based gel foods not only to improve their textural and physical properties, but also to add to their health benefits.

## 5. Conclusions

In this study, okara was atomized using a WG system to produce NCs. We treated okara with a WG system at a varying number of passages. The median PSD size decreased with an increase in the WG treatment passages up to the third time and was reduced to 8.9 μm by the fifth time. The viscosity and dispersion performance of okara 24 h post WG-treatment increased with an increase in the number of passages. These results demonstrated that the physicochemical properties of okara could be enhanced by using the WG system. 

The rheological properties and WHC of SPI gels improved upon the addition of WG-treated okara at 1.0% and 1.5% NaCl concentrations, which are used in surimi and processed meat products. This effect was increased with an increase in the concentration of WG-treated okara. The present study demonstrated that the WG-treated okara could be used for the development of protein gel-based foods to enhance gel properties.

## Figures and Tables

**Figure 1 foods-10-00348-f001:**
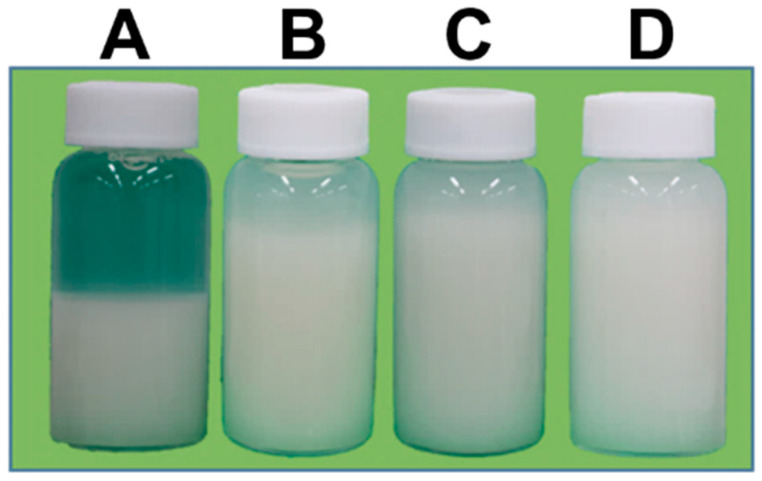
Wet-type grinder (WG)-treated okara in water after 24 h. Okara (2 wt%) was treated with the WG for different passages. (**A**) untreated; (**B**) one passage; (**C**) three passages; (**D**) five passages.

**Figure 2 foods-10-00348-f002:**
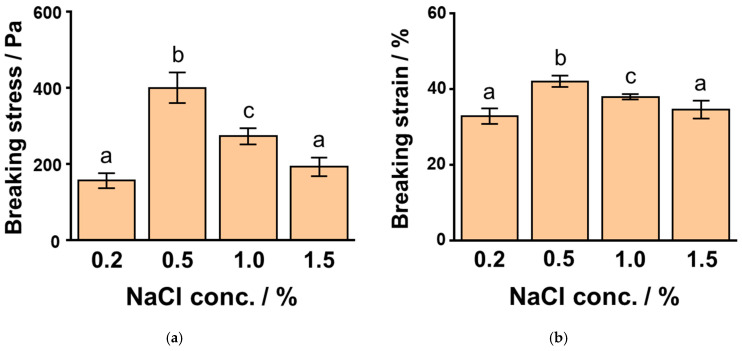
The breaking stress (**a**) and strain (**b**) of 6% soybean protein isolate (SPI) gels at different concentrations of sodium chloride (NaCl). Different letters indicate significant differences (*p* < 0.01).

**Figure 3 foods-10-00348-f003:**
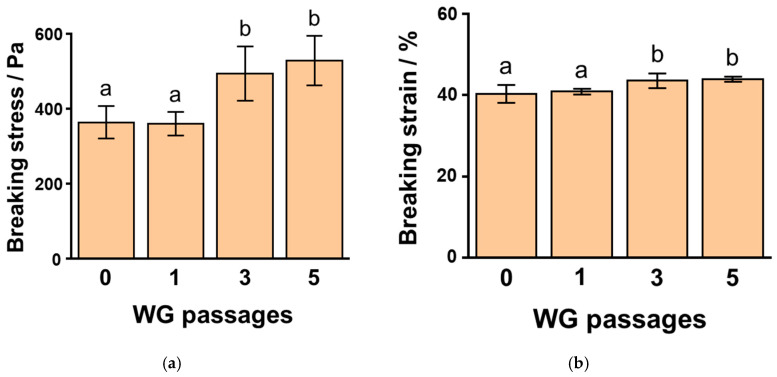
The breaking stress (**a**,**c**) and strain (**b**,**d**) of 6% soybean protein isolate (SPI) gels to which 1% wet-type grinder (WG)-treated okara was added after different passages at 1.0% (**a**,**b**) and 1.5% (**c**,**d**) sodium chloride concentration. Different letters indicate significant differences (*p* < 0.01).

**Figure 4 foods-10-00348-f004:**
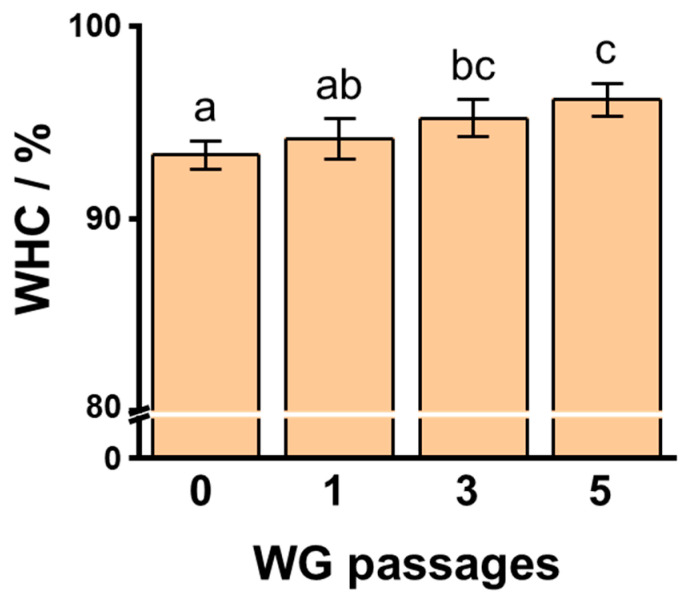
The water holding capacity (WHC) of 6% soybean protein isolate (SPI) gels to which 1% wet-type grinder (WG)-treated okara was added after different passages at 1.0% sodium chloride concentration. Different letters indicate significant differences (*p* < 0.01).

**Figure 5 foods-10-00348-f005:**
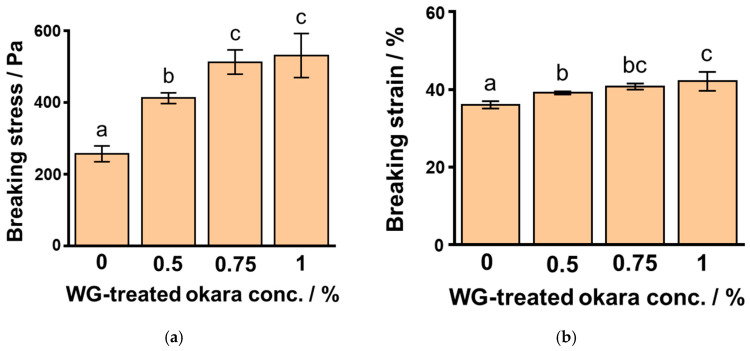
The breaking stress (**a**) and strain (**b**) of 6% soybean protein isolate (SPI) gels at 1.0% NaCl with the addition of five-time wet-type grinder-treated okara at different concentrations. Different letters indicate significant differences (*p* < 0.01).

**Table 1 foods-10-00348-t001:** Median sizes from particle size distributions and viscosities of wet-type grinder-treated okara (2 wt%) at a shear rate of 0.5 s^−1^ after different passages (25 °C).

	Untreated	One Passage	Three Passages	Five Passages
Median size (μm)	68.5	13.5	9.9	8.9
Viscosity (mPas)	10 ± 4 ^a^	40 ± 8 ^b^	70 ± 10 ^c^	120 ± 16 ^d^

Different letters indicate significant differences (*p* < 0.05).

## Data Availability

Not applicable.

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
