# Peer review of "Developing Soybean Protein Gel-Based Foods from Okara Using the Wet-Type Grinder Method"

_foods, 2021, doi:10.3390/foods10020348_

Round 1

Reviewer 1 Report

15. Change the writing style of "we treated" to "okara was treated" etc. 

State the specific objectives of the study at the end of the introduction. 

Materials and methods - Explain the reasons for using different time-temperature relationships at different methods or refer back to previous literature where these methods were used. 

150. "We treated" doesn't sound that scientific. Change this style of writing. 

Table 1 - Conduct analysis to see of there is a significant difference between treatments. Apply for all results 

Author Response

Reviewer 1

Thank you for your valuable comments on our manuscript. We believe that our manuscript has been improved by your comments. The revised manuscript with all the modifications was marked in red color. Please see my responses to your comments below.

  1. Change the writing style of "we treated" to "okara was treated" etc.

Response: I changed the writing style (L16).

State the specific objectives of the study at the end of the introduction.

Response: I added the specific objectives at the end of the introduction (L82-86)

Materials and methods - Explain the reasons for using different time-temperature relationships at different methods or refer back to previous literature where these methods were used.

Response: I added the reasons for using different time-temperature relationships and references where the methods were used in 2.5 (L108) and 2.7 (L123-125).

  1. "We treated" doesn't sound that scientific. Change this style of writing.

Response: I changed the writing style (L146)

Table 1 - Conduct analysis to see of there is a significant difference between treatments. Apply for all results

Response: We measured three times for viscosity and two times for particle size distribution (PSD). Thus, I statistically analyzed viscosity but not PSD (Table 1).

Reviewer 2 Report

The paper deals with an interesting topic appropriate for the Journal's aims. The manuscript is written at a good level, adequately divided into individual chapters. The abstract is concise, the literature search is focused on the topic, the data are clearly presented and adequately commented in the discussion. I have only a minor suggestion, that in my opinion could improve the paper impact. The introduction section is very comprehensive, discusses the properties of okara in great detail but could have been more concise in some sections – result and discussion should be in the foreground of the research article.

Author Response

Reviewer 2

Thank you for your valuable comments on our manuscript. The revised manuscript with all the modifications were marked in red color. We believe that our manuscript has been improved by your comments. Please see my responses to your comments below.

Comment: I have only a minor suggestion, that in my opinion could improve the paper impact. The introduction section is very comprehensive, discusses the properties of okara in great detail

but could have been more concise in some sections – result and discussion should be in the foreground of the research article.

Response:

Results: I add explanations to be clear why we did (L145-146, L162-164, L175-176, and L210-211).

Discussion: I deleted the sentence “such as imitation crab legs and sausages”.

Reviewer 3 Report

This article has good organization and explanations. The benefits of nanocellulose and okara are explained. The objectives are listed: Using wet-type grinder for okara atomization and investigating the effects of WG treated okara on soybean protein isolates. 

The results suggest WG can improve the physiochemical properties of okara and WG treated okara improved SPI gel properties. 

Areas of concern:

  1. The data in-text for figure 5a does not seem to match the graph. 
  2. Reference section: 
    1. 6. capital P in Pickering is not consistent
    2. 10. capital letters in title
    3. 18. Capital letter in title
    4. 25. Capital letter in title
    5. 30. No period at the end
    6. 35. Capital letter in title

Author Response

Reviewer 3

Thank you for your valuable comments on our manuscript. We believe that our manuscript has been improved by your comments. Please see my responses to your comments below.

1. The data in-text for figure 5a does not seem to match the graph.

Response: I revised figure 5a.

2. Reference section:

Response: I changed the order of references that 6, 10, 18, 25, 30 and 35 are 15, 19, 3, 10, 27, and 30, respectively.

1) 6. capital P in Pickering is not consistent

Response: Ref. 15, Capital P is not wrong because Pickering is name.

2) 10. capital letters in title

Response: Ref. 19, I corrected capital letters in the title.

3) 18. Capital letter in title

Response: Ref. 3, I corrected capital letters in the title.

4) 25. Capital letter in title

Response: Ref. 10, I corrected O to o in the title.

5) 30. No period at the end

Response: Ref. 27, I added period at the end.

6) 35. Capital letter in title

Response: Ref. 30, I corrected V to v in the title.

Reviewer 4 Report

The article “Developing Soybean Protein Gel-Based Foods from Okara using the Wet-Type Grinder Method is good written and need to enhance on multiple positions.

Abstract/Introduction:

The authors lack to develop and outline a clear Hypothesis and a working hypothesis the literature is nicely reported but authors need to reflect to it and conclude a hypothesis from where the planed necessary work is planned to proof the hypothesis. Provide evidence that others do not have done so fare and the gar of knowledge.

The introduction needed to be reworked/reorganized.

For the introduction: start with: Okara explanation (now at 63) followed by the size reduction (waterjet) followed by NC (what is now at the start)

Reduce the part standing in line 40-52 by half!

Make clearer why the SPI was chosen! As a good model to show the effects. Line 88-95 is not good chosen to explain it no body eat a SPI gel!!

Line 29: does the NC has these properties? I would believe more the products that are produced with the NC.

Line 31: not waste! Better side steams

Line 34: atomize: this term has several meanings: do you mean size reduction? Then please use a more appropriate term.  Like in Line 104??

Line 59/60: this statement doe not needed here.

Line 61/62: this statement is to early! Later at the end of the introduction together with the hypothesis and assessment of the hypothesis.

line 77/78: can be deleted this statement from the citation does not fit here.

Line 102 a clear specification of the product is missing and need to provide to the reader!

Line 130 and the following: more information is necessary; it is not possible to reproduce your work with the provide information.

Line 150 state the different passage directly

Line 153 repetition

Mat & Methods: make a clear statement about you repetition of your measurements and on your experiments provide more information!!!

Results

Line 162/Figure 2 it does not make sense to show this alone, furthermore, what is the benefit form show same results in % and Pa? Provide new figures: You can than combine figure 5a and 6a and accordingly for the others, Figure 4a and 3a!

In Figure 2: significant? In general, choose a more common way to show the significant

Figures: And overall authors can do one graph to compare which factor of the tested hast the biggest influence on the breaking stress.

Round 2

Reviewer 4 Report

thank you for enhancing the manuscript!

still there are major change to do to higher the quality...

line 34,39,40,41 and 67: Please do not use the term they,.. authors can be name, and maschines products etc. aswell

Please provide a research aim

line 108 how often where the whole experimental setup repeated add this information

still you need to modify the graphs, the presentation is week

  1. please put the breaking strain of all graphs in a table! The bar chart does not help to illustrate.
  2. Figure 3 and 5 can be combined to one graph with each two (two salt concentration) bars for the number of passages. and a third bar with a second y-axis with than indicate the water holding capacity.
  3. Figure 2 and 6 can be set up in one 3d plot with tow influences the concentration of salt and okara.

Author Response

Reviewer 4

Thank you for your valuable comments on our manuscript. We believe that our manuscript has been improved by your comments. The revised manuscript with all the modifications was marked in red color. Please see my responses to your comments below.

Comment: line 34,39,40,41 and 67: Please do not use the term they,.. authors can be name, and maschines products etc. aswell

Response: I revised that I did not use the term they (L34, L39-42)

Comment: Please provide a research aim

Response: I added the research aim (L84-85).

Comment: line 108 how often where the whole experimental setup repeated add this information

Response: I described the method of preparation of SPI gels but not  measurements in line 108. Thus, I cannot state how often where the whole experimental setup repeated.

Comment: still you need to modify the graphs, the presentation is week. please put the breaking strain of all graphs in a table! The bar chart does not help to illustrate.

Response: I indicated the values of breaking strain in the text (L166-167, L180-181, L193-194, L212-213).

Comment: Figure 3 and 5 can be combined to one graph with each two (two salt concentration) bars for the number of passages. and a third bar with a second y-axis with than indicate the water holding capacity.

Response: I combined Fig. 3 and 5 and showed as Fig. 3.

Comment: Figure 2 and 6 can be set up in one 3d plot with tow influences the concentration of salt and okara.

Response: Fig. 2 shows the breaking stress and strain of SPI gels at different NaCl concentration without WG-treated okara, while Fig. 6 (Fig. 5 in the revised manuscript) indicates the breaking stress and strain of SPI gels at 1.0% NaCl with adding WG-treated okara at different concentrations. Thus, I cannot set up these two graphs as you advised.